

# A new tracking algorithm for sea ice age distribution estimation

Anton Andreevich Korosov[1], Pierre Rampal[1], Leif Toudal Pedersen[2], Roberto Saldo[2], Yufang Ye[3], Georg Heygster[4], Thomas Lavergne[5], Signe Aaboe[6], and Fanny Girard-Ardhuin[7]

[1]Nansen Environmental and Remote Sensing Center, Thormohlensgate 47, Bergen, 5009, Norway
[2]Technical University of Denmark, Lyngby, Denmark
[3]Chalmers University of Technology, Gothenburg, Sweden
[4]University of Bremen, Bremen, Germany
[5]Norwegian Meteorological Institute, Oslo, Norway
[6]Norwegian Meteorological Institute, Tromsø, Norway
[7] Ifremer, Univ. Brest, CNRS, IRD, Laboratoire d'Oceanographie Physique et Spatiale (LOPS), IUEM, 29280, Brest, France

*Correspondence to:* Anton Korosov (anton.korosov@nersc.no)

**Abstract.** A new algorithm for estimating sea ice age (SIA) distribution based on the Eulerian advection scheme is presented. The advection scheme accounts for the observed divergence/convergence and freezing/melting of sea ice and predicts consequent generation/loss of new ice. The algorithm uses daily gridded sea ice drift and sea ice concentration products from the Ocean and Sea Ice Satellite Application Facility and produces gridded fractions of sea ice of a specific age. Thus, each grid cell of the output product contains a frequency distribution of sea ice age allowing to apply mean, median, weighted average or other statistical measures. The produced SIA maps and time series are compared to the National Snow and Ice Data Center SIA product. Several improvements related to the usage of the new ice drift product, constraining the algorithm by the observed ice concentration and preventing undersampling by the Eulerian scheme are demonstrated. Muliyear ice (MYI) concentration is computed as a sum of all multi-year ice fractions and compared to the MYI products based on passive and active microwave and SAR products.

## 1 Introduction

Sea Ice Age (SIA) is one of the components of the Essential Climate Variable (ECV) for sea ice as defined by the Global Climate Observing System (GCOS) (WMO, 2015). It is an important climate change indicator in polar regions, which describes the sea ice cover state in addition to its concentration and thickness. More generally, the changes in sea ice age can be seen as a proxy of sea ice decline in Arctic.

Numerous studies have been focusing on the estimation and evolution of the Arctic sea ice age over the last decades (Fowler et al., 2004; Rigor and Wallace, 2004; Maslanik et al., 2007; Tschudi et al., 2010; Maslanik et al., 2011; Tschudi et al., 2016c). They all report that the amount of old/thick ice in the Arctic Ocean has been decreasing dramatically over the last decades.



For example, the very old ice (> 4 years old) comprised 20% of the ice pack in 1985, decreasing to 3% in 2015, while the 3-4 years old ice decreased by one third over the same time period (Perovich et al., 2015). Kwok (2004), Polyakov et al. (2012) and Kwok and Cunningham (2015) reported significant decline in the area covered by Multi-Year ice (MYI) in the central Arctic since 1999, to which they associated a significant decrease in the mean sea ice thickness and volume in the same region.

Sea ice age is an important parameter of the Arctic ocean system and may be a good indicator of significant changes in the dynamical and thermodynamical regimes that have taken place since the beginning of the current century such as sea ice thinning and faster ice drift. The younger –seasonal ice cover present in the Arctic nowadays is in general thinner, which makes it more vulnerable to break up, deform and drift under the actions of waves, winds and currents. This has been confirmed by calculating the trends in sea ice drift and deformation over the last 3 decades, which are significant and respectively about

13% and 50% per decade (Rampal et al., 2009). If more mobile, sea ice might also be exported more easily and rapidly out of the Arctic basin, e.g. through Fram Strait, especially the MY sea ice pack located north of Greenland and Ellesmere Islands. A more fragmented sea ice cover is also more vulnerable to summer melt. For example, during the winter of 2007 – 2008 large chunks of MYI resulting from the intense break up of the thickest part of the Arctic ice cover located in the north of the Canadian Archipelago were melted away while drifting in the Beaufort Sea within the few following summer months (Kwok

and Cunningham, 2010). A positive feedback on the loss of MYI due to the more frequent sea ice cover fracturing, increased sea ice drift and enhanced melting rate is therefore potentially activated, contributing to the observed negative trend in sea ice age.

It is the purpose of the present paper to describe a method and a derived dataset that allow us to shed more light on the development of the age distribution of the Arctic sea ice. For this purpose, we have taken advantage of some new datasets on

sea ice drift and concentration developed and distributed by the EUMETSAT Ocean and Sea Ice Satellite Application Facility (OSI SAF). In addition, we have developed a new Eulerian scheme of advection supported by the Sea Ice Climate Change Initiative (SICCI) project of the European Space Agency (ESA). These improvements have allowed us to produce a new sea ice age dataset which in each grid box contains not only the age of the oldest ice, but the actual age distribution provided as fractions of ice of different age categories (hereafter refered to as *sea ice age fractions*). The dataset will be presented and

compared with earlier attempts to map Arctic sea ice age as well as with the standard products for sea ice type classification from scatterometer and microwave radiometer observations.

## 2   Data

### 2.1   Sea ice drift

Information on sea ice motion was acquired from two sources. First, the National Snow and Ice Data Center (NSIDC) sea ice

drift (SID) product v.0116 (Tschudi et al., 2016c) was downloaded from the NSIDC portal (Tschudi et al., 2016b). Weekly SID fields from NSIDC were accessed on an Equal Area Scalable Earth (EASE) grid (Brodzik et al., 2012) with 25 km spacing for the period from October 1978 to December 2015. As pointed out by (Szanyi et al., 2016) this ice drift product contains artifacts due to the composition of the ice drift derived from satellite data, observed by *in situ* buoys and predicted using a simple free



drift model. On short time scales, these artifacts result in large openings in the predicted distribution of multi-year ice (MYI) which are filled with the first-year ice (FYI). On longer time scales the openings get bigger and the shape of the predicted ice age distribution may be significantly distorted.

The second SID product was produced by the Ocean and Sea Ice Satellite Application Facility (OSI SAF) High Latitude
Processing Center. The low-resolution sea ice drift product from the EUMETSAT OSI SAF is operational since 2009. It implements the Continuous Maximum Cross-Correlation algorithm of Lavergne et al. (2010) for retrieving ice drift vectors from daily composited maps of various medium resolution passive and active microwave (PAMW) satellite sensors such as the SSMIS, AMSR2, and ASCAT. A multi-sensor analysis is also provided (Lavergne, 2016a). The retrieval of SIA requires sea ice drift information during summer. This was recently achieved by the OSI SAF product (Lavergne, 2016b) using the 18.7 GHz
channels of the GCOM-W1 AMSR2 instrument (Kwok, 2008). For this study, a short re-processed record of OSI SAF ice drift product (Oct 2012 - May 2017) was accessed, complemented by the operational product available from http://osisaf.met.no (May 2017 onwards). The SID product from OSI SAF was filtered component-wise with 3 x 3 pixels median filter and then upscaled using linear interpolation onto a grid with 10 km spatial resolution. Gaps in the product were filled with nearest neighbor values.

## 2.2   Sea ice concentration

Sea Ice Concentration (SIC) product v.1.6 (Tonboe et al., 2017a) was produced by the OSI SAF portal at 1 day temporal resolution and 10 km spatial resolution for the period September 2012 - September 2017 covering the Northern Hemisphere. Validation of the product indicates that SIC can be retrieved with 10% accuracy with slight seasonal variations (Tonboe et al., 2017b).

## 2.3   Sea ice type (MYI concentration)

Sea ice types can be discriminated with PAMW satellite observations since the physical signatures of sea ice change significantly after the influence of summer melt and brine rejection. Therefore sea ice that has survived at least one summer melt is referred to as multiyear ice, and seasonal ice is referred to as first-year ice.

The algorithm of Environment Canada Ice Concentration Extractor (ECICE) (Shokr et al., 2008) can combine several dif-
ferent observation inputs, e.g. passive and active microwave satellite data. Combined PAMW data can help to identify MYI, however, the retrieval shows flaws under specific weather conditions. The improved MYI concentration retrievals developed at University of Bremen is based on the ECICE algorithm using brightness temperatures from AMSR2 and radar backscatter from C-band scatterometer ASCAT. The initial MYI concentrations are corrected by two correction schemes: one using temperature records from atmospheric reanalysis to identify MYI anomalies caused by warm spells and replace with interpolated
MYI concentrations (Ye et al., 2016a), and the other utilizing mainly ice drift records to constrain the MYI changes within a plausible contour (Ye et al., 2016b).

The improved MYI concentrations were provided as gridded products on polar stereographic grid with 12.5 km spatial resolution for the winter months (November - April) of 2013-2017. Compared to the MYI dataset that the corrections were





developed to apply on, the retrievals from AMSR2 and ASCAT are much coarser (12.5 km vs 4.45 km). The coarser resolution requires adjustment of the thresholds in the drift correction, which results in suboptimal performance of the drift correction, therefore could lead to unexpected problems in the final dataset.

The OSI SAF sea ice type is another algorithm that combines both passive and active microwave data in a Bayesian approach (Aaboe et al., 2017) that computes the probability of occurrence of the most likely ice type - first-year ice or multiyear ice. The OSI SAF type product is a near-real-time product that has been operational since 2005 with data available from the OSI SAF High Latitude processing centre [http://osisaf.met.no]. Since the start of operational production the algorithm has been upgraded several times by including new sensors and improving the methods. For the period of interest in this study, 2013-2017 summer, the retrieval is based on scatterometer data from the ASCAT instrument and atmospherically corrected brightness temperatures from SSMIS. From October 2015 an algorithm upgrade resulted in the implementation of dynamical training data updated on a daily basis in order to replace the previous fixed training data. The ice type product is provided for the winter period October until mid-May on polar stereographic projection with 10 km grid resolution.

## 2.4 Sea ice age

Information on sea ice age (SIA) independent of the method introduced here was acquired from the NSIDC portal (Tschudi et al., 2016a). Weekly SIA fields were provided at EASE grid (Brodzik et al., 2012) with 12.5 km spatial resolution for the period from October 1978 to December 2015. The gridded SIA product is generated from positions of the virtual Lagrangian ice parcels (initially released on a regular grid) (Fowler et al., 2004) when the grid cell is assigned the age of the oldest parcel. This does not take into account fractions of younger ice present in the same cell. It may happen that most of the drifters within a cell are very young and only one drifter is old but nonetheless the entire cell is still assigned to be the oldest ice. Such case is shown on Fig. 1 for 1 Jan 1985. For generating this figure we have implemented the NSIDC ice age algorithm and applied it to the NSIDC ice drift product starting from 1978. Concentration of each ice age category is calculated as a relative number of ice parcels with respective age.





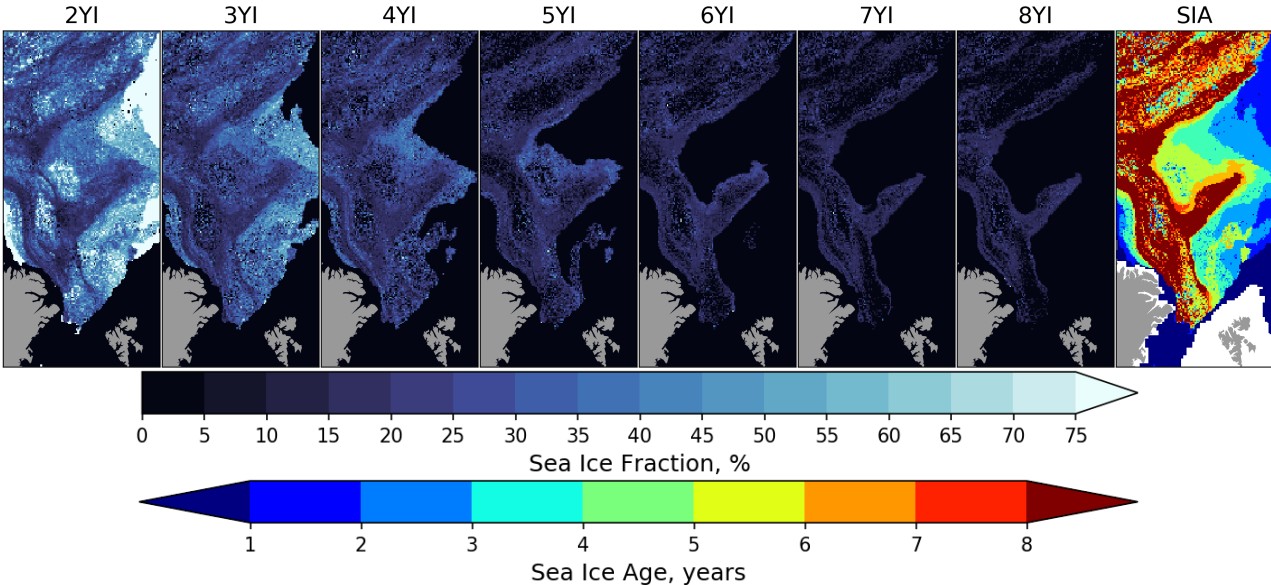

**Figure 1.** Sea ice age fractions for 1 Jan 1985 in the Greenland and Lincoln Seas. It is clearly seen that in many areas the fraction of 8YI does not exceed 20%, however the corresponding SIA map shows that the entire pixel is assigned to contain 8YI.

## 3 Methodology

The sea ice age computation is implemented in two stages as explained in details below. First, sea ice age fractions are independently advected using the satellite derived sea ice drift and concentration observations and the Eulerian advection scheme. Second, the sea ice age is computed accounting for concentration of each ice age fraction.

### 3.1 Eulerian advection scheme

At each time step the observations of sea ice drift velocity components ($U$ and $V$) and sea ice concentration ($C_{OBS}$) are provided as gridded products on the polar stereographic grid with the same spatial resolution with pixel size in $x$ and $y$ directions equal to $R_x$ and $R_y$, correspondingly. We assume that a sample ice parcel in a pixel has shape of one grid cell and an initial ice concentration $C$ and corresponding velocity components $U$ and $V$. An example is presented on Fig. 2, where the ice parcel originating from the pixel 31 is shown as a blue square. The parcel drifts from point $A$ over time $\Delta t$ and the coordinates of the destination point $B$ can be computed as follows:

$$X_B = X_A + U\Delta t$$
$$Y_B = Y_A + V\Delta t \tag{1}$$



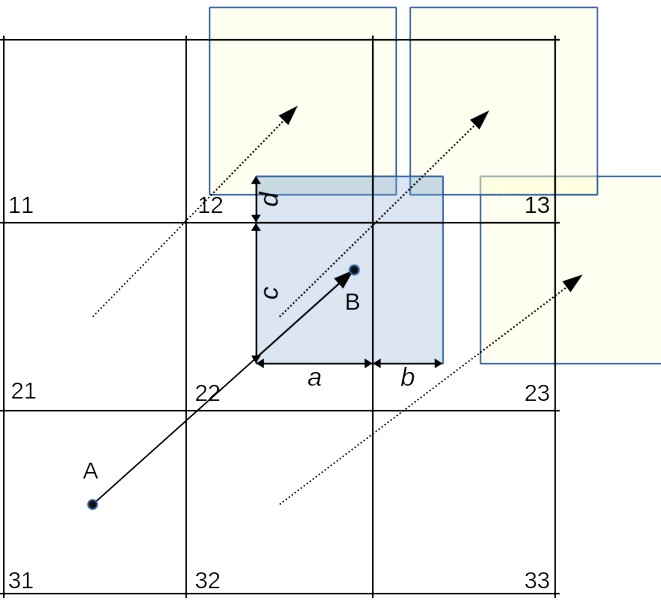

**Figure 2.** Scheme of Eulerian advection of sea ice. Points A and B denote start and end of drift of an ice parcel shown by the blue square.

The areal fluxes of sea ice out of the donor pixel 31 into recipient pixels 12, 13, 22 and 23 are calculated as follows:

$$F_{31 \blacktriangleright 12} = C_{31} a d$$
$$F_{31 \blacktriangleright 13} = C_{31} b d$$
$$F_{31 \blacktriangleright 22} = C_{31} a c \tag{2}$$
$$F_{31 \blacktriangleright 23} = C_{31} b c$$

where $a$, $b$, $c$ and $d$ are sides of the rectangles formed by intersection of the displaced donor pixel and recipient pixels (Fig. 2). The recipient pixels are selected based on position of point $B$ - the four closest pixels become recipients. Concentration of sea ice at the next step in a recipient pixel (e.g. pixel 13 on Fig. 2) ($C_{13}^\star$) is calculated as:

$$C_{13}^\star = F_{21 \blacktriangleright 13} + F_{22 \blacktriangleright 13} + F_{31 \blacktriangleright 13} + F_{32 \blacktriangleright 13} \tag{3}$$

or in generalized form:

$$C_R^\star = \sum_{4 \, donors} C_D (1 - \frac{|X_R - X_D - U \Delta t|}{R_x}) \, (1 - \frac{|Y_R - Y_D - V \Delta t|}{R_y}) \tag{4}$$

where $C_D$ is the concentration of a donor pixel, $X_D$ and $Y_D$ are the coordinates of a donor pixel, $X_R$ and $Y_R$ are the coordinates of the recipient pixels and $R_x$, $R_y$ - pixel size.

When the observed sea ice drift field diverges then a gap appears between the ice parcels (e.g. in pixel 23). If the sum of fluxes (eq. 4) into a recipient pixel falls behind the actually observed concentration then this is interpreted as opening of leads, freezing of water in the leads and generation of new ice. Then, ice is presented by two fractions: the partial concentration of





the older ice $C_{OI}$ is computed as the sum of incoming fluxes (eq. 4) and the partial concentration of the young ice $C_{YI}$ is computed as a remainder:

$$C_{YI} = C_{OBS} - C_{OI} \tag{5}$$

When the observed sea ice drift field converges then the advected ice parcels overlap (e.g. in the pixel 12) and the sum of fluxes into a recipient pixel may exceed the concentration observed by satellites. This is interpreted as generation of pressure ridges and increase in thickness of the older fraction of sea ice. In that case the total ice concentration at the next step is assigned to be the observed ice concentration.

During the freeze-up period the observed concentration increases and becomes higher than the sum of fluxes into a recipient pixel. This is also interpreted as generation of YI only and eq. 5 is used for computing YI concentration. During melting the observed concentration decreases and it may become less than the total predicted concentration (sum of OI and YI fractions). In that case the YI concentration is decreased first, and if YI is absent then the OI concentration is decreased.

## 3.2 Advection of sea ice age fractions

Advection of a sea ice age fraction is initiated on 10 September, the approximate date when ice extent reaches minimum in the Arctic, area of first-year ice (FYI) is zero and all observed sea ice is multi-year ice (MYI) by definition. The age of each ice fraction is increased by one year also on 10 September of each consecutive year of advection.

In our study we initiated SIA calculation on 1 October 2012 when continuous high quality observations of ice drift started to be available from AMSR2 (Fig. 3). Propagation of ice age fractions from later years started from 10 September.

Since we did not know the spatial distribution of ice of different age within the pack at the first moment of time (1 Oct 2012), we postulated that all observed ice is second year ice (2YI). Or, in other words, the concentration of second year ice on 1 Oct 2012 is assigned to be equal to the total observed concentration: $C_{2YI} = C_{TOT}$ (Fig. 3, A). Then the developed advection scheme is applied on a daily basis utilizing the available daily ice drift and concentration products (Fig. 3, B - E). From 1 Oct 2012 to 10 September 2013 (Fig. 3, A - D) the advected fraction contains second year ice (2YI) but on 10 September 2013 the age of this fraction is increased by 1 year and later it becomes the fraction of the third year ice - 3YI (Fig. 3, E).





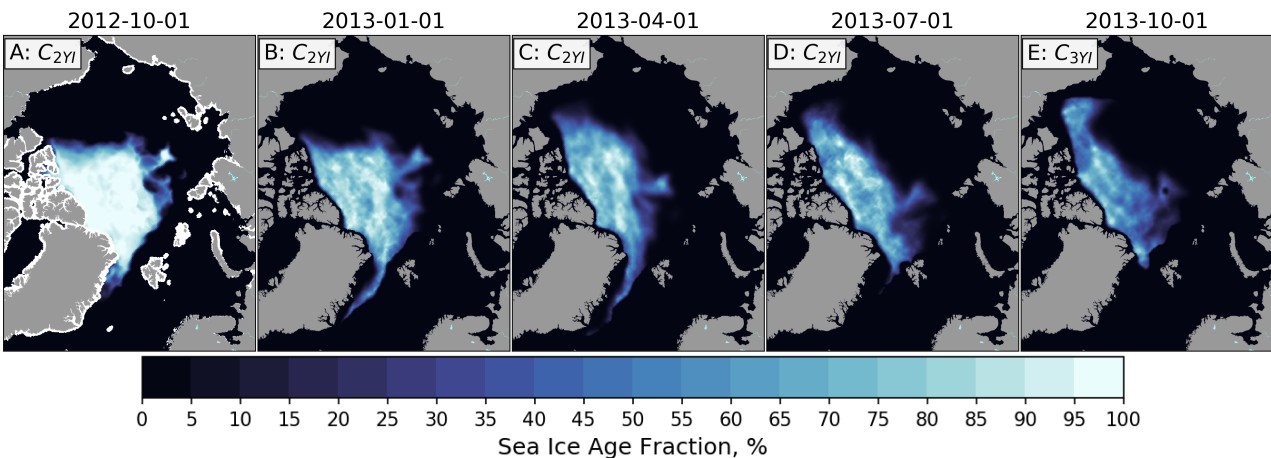

**Figure 3.** Example of advection of an ice fraction from 1 Oct 2012 to 1 Oct 2013 shown for every third month.

On 10 Sep 2013 the total ice concentration is the sum of all multiyear ice fractions and therefore the initial concentration of the second year ice is calculated as:

$$C_{2YI} = C_{TOT} - C_{3YI} \qquad (6)$$

where $C_{TOT}$ is the ice concentration observed by satellites and $C_{3YI}$ is the ice fraction advected from 1 Oct 2012. Both 2YI and 3YI fractions are advected further using the developed scheme and after $N$ years of advection on each of 10 Septembers the concentration of 2YI is calculated as:

$$C_{2YI} = C_{TOT} - \sum_i C_i \qquad (7)$$

where $C_i$ - is the concentration of the ice fraction advected from previous years.

### 3.3 Computation of sea ice age

At any moment of time, sea ice in each cell is characterized by a vector of concentrations of ice fractions of various age with the sum equal to the total sea ice concentration (Fig. 4). In other words the new algorithm provides ice age probability distribution. A single number that characterizes a probability density function can be computed in several standard ways: minimum, maximum, mean, median, mode, etc. We propose that sea ice age can also be computed from the concentration of ice age fractions using several approaches depending on the preferred definition of the ice age (Fig. 5):

– **Maximum age**: Age of the oldest ice fraction that exceeds a given threshold (e.g. 5% concentration).

– **Average age**: Mean of age of the ice fraction that exceeds a given threshold.

– **Modal age**: Age of the ice fraction that has the highest concentration.




- **Median age**: Age of the ice fraction that splits the age distribution into two equal parts. Linear interpolation between the ice age fractions or approximation of ice distribution by a predefined function may be needed to compute median age correctly. More generally, median (the value of the $50^{th}$ percentile) can be replaced by any percentile.

- **Weighted average age**: Average of ages of individual ice fractions weighted by their concentrations:

$$A = \frac{\sum_f A_f C_f}{\sum_f C_f} \tag{8}$$

where $A_f$ - age of a sea ice fraction, $C_f$ - concentration of a sea ice fraction.

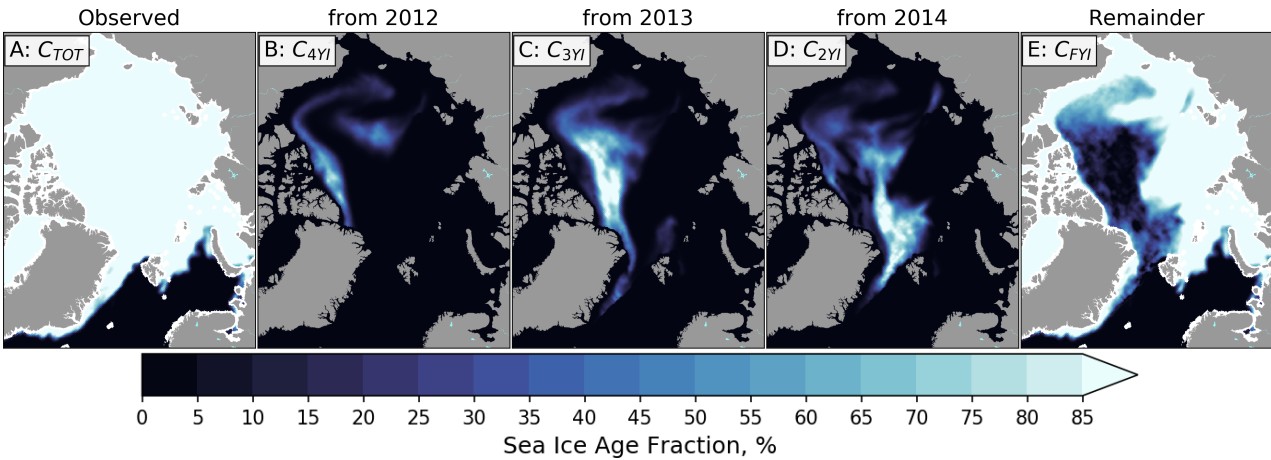

**Figure 4.** Maps of sea ice age fractions on 1 March 2015. (A) observed total sea ice concentration, (B, C, D) fractions of advected multiyear ice, (E) FYI fraction. Colorbar represents the sea ice age fraction concentration.

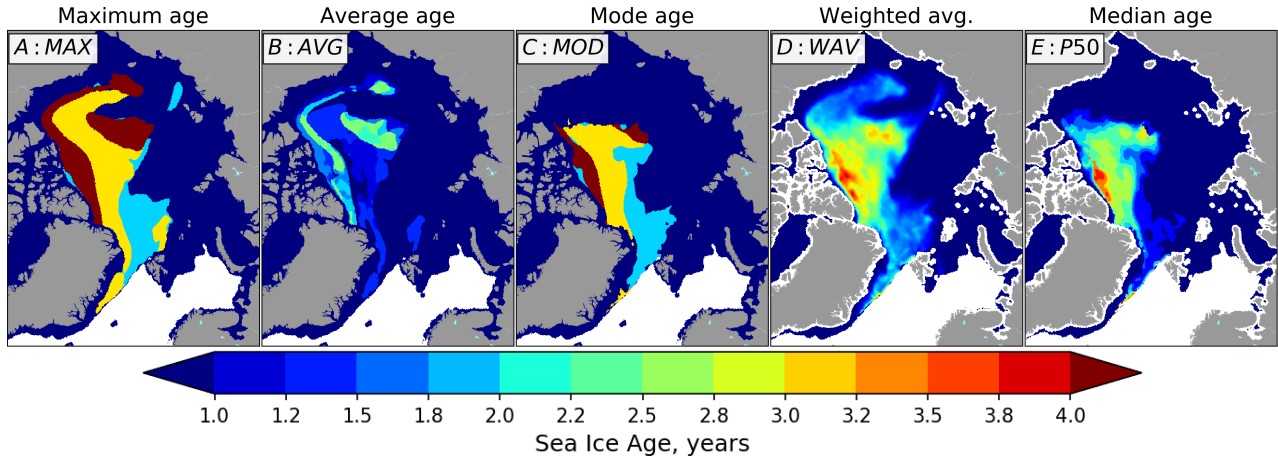

**Figure 5.** Maps of sea ice age on 1 March 2015 computed with different methods. (A) maximum ice age, (B) average ice age, (C) modal ice age, (D) median (percentile 50) ice age, (E) weighted average ice age. Colorbar represents the sea ice age in years.





## 4  Results

The developed advection scheme was applied to the sea ice drift and concentration products available at OSI SAF from
October 2012 to 1 March 2017. We compared our products to the already available sea ice products including SIA maps
from NSIDC; time series and maps of MYI from NSIDC, Bremen University, OSI SAF and DTU. The results of comparison
presented below can be used for indirect assessment of our SIA algorithm.

### 4.1  Comparison with sea ice age product from NSIDC

In order to provide stepwise illustration of the SIA product improvements we have used four different combinations of forcings
and algorithms to produce a SIA map for the 1 Jan 2016. First, we have implemented the NSIDC advection and SIA algorithms
and applied it to the ice drift product from NSIDC. Second, we have applied the NSIDC algorithm to the ice drift from OSI
SAF. Third, we have applied our Eulerian advection scheme and SIA algorithm to the ice drift from OSI SAF but without
accurately accounting for SIC. In this experiment all pixels with SIC below 15% were assigned to be open water and other
pixels contain 100% ice. Finally, we have used the new advection scheme, the OSI SAF ice drift and fully accounting for SIC.

Comparison of the generated ice age products is presented on (Fig. 6). The panels A and B contain maps of maximum
age computed with the NSIDC algorithm and panels C and D contain ice age estimated with the weighted average. The
comparison indicates that spatial distributions of SIA are similar across four combinations of forcings and algorithms only in
general. The change of the forcing significantly affects distribution of SIA (Fig. 6, A and B). The belt of very old sea ice in the
Lincoln Sea almost disappears, large areas with old ice in the Central Arctic become more pronounced, long sleeves of 2YI
with interspersing of older ice stretch across Central Arctic towards the Laptev Sea and along the Beaufort sea coast into the
Chukchi sea, vast areas of 2YI in the European Arctic become more homogeneous.





**Figure 6.** Comparison of SIA for the 1 Jan 2016 calculated with the following combinations of forcings and algorithms: A) NSIDC drift, NSIDC algorithm; B) OSI SAF drift, NSIDC algorithm; C) OSI SAF drift, SICCI algorithm without SIC; D) OSI SAF drift, SICCI algorithm with SIC.





The change of the advection scheme does not affect the overall spatial distribution of MYI significantly (Fig. 6, B and C) but generates a much smoother picture without speckles. Using the weighted average for SIA computation dramatically decreases the observed age (Fig. 6, C). Few small spots where SIA reaches 4.5 - 5 years are observed only near the Canadian Archipelago and in the Central Arctic. A stripe of high SIA appears along sea ice edge in the Fram Strait. This ice was advected from the

Central Arctic but fractions of younger ice have melted and only fractions of older ice remain making the weighted average value to become high.

When the new algorithm takes SIC into account then SIA is decreasing even more and the ice older than 4 years is observed only near the Canadian Archipelago coast (Fig. 6, D). Moreover, extent of MYI is also decreased and the sleeves of 2YI towards the Laptev and Chukchi seas almost disappear.

## 4.2 Intercomparison of multi-year ice concentration products

Multi-year ice (MYI) concentration is an important indicator of abundance of older, thicker and rougher sea ice and is readily available from several resources including OSI SAF (Aaboe et al., 2017) and University of Bremen (Ye et al., 2016a). MYI can also be estimated from the NSIDC SIA product if we assume that bins with ice older than 1 year contain 100% MYI and other bins contain 0%. Total concentration of MYI from the SICCI SIA product was estimated as sum of all fractions of MYI.

Intercomparison of MYI maps from four sources indicated that the general spatial distribution and temporal evolution is rather similar (Fig. 7): most of MYI is observed in the Canadian Sector, a thin filament is exported through the Fram Strait, in some years the Beaufort Gyre advects MYI along the Beaufort Sea coast, sleeves of enhanced MYI concentration are elongated towards the Laptev Sea.

Distinct features of MYI from NSIDC (Fig. 7, first row) include the following: the map is binary (ice / no ice); numerous

speckles are present; rather large lacunas filled mostly with FYI appear in the ice pack (e.g. 2015-12-31, Lincoln Sea); the MYI sleeves that extend towards the Laptev Sea are rather wide, a wide gap between ice pack and the coastline is observed. The SICCI product (Fig. 7, second row) is continuous and smooth, expectedly it shows higher MYI concentrations in the Central Arctic and near the Canadian coast. The OSI SAF MYI product (Fig. 7, third row) is also binary but the map is not speckly, generally it shows lower extent of MYI than other products, but in some period the extent is much higher (e.g. on 2017-03-29

in the Beaufort Sea). The UB product is also continuous and apparently can resolve smaller spatial details but the atmospheric influence seem to reduce MYI concentration in the middle of the ice pack (e.g. along the 150° meridian in 2013 or above the Yermak plateau in 2015).



**Figure 7.** Comparison of MYI concentration estimated from the NSIDC SIA product (upper row), from the SICCI SIA product (second row), downloaded from OSI SAF (third row) and from University of Bremen (fourth row). Daily maps are generated for 31 December of the years from 2012 to 2016 and for 29 March 2017 from available data. Red contours denote extent of MYI in the SICCI product at 5% concentration.

The time series of MYI area are compared both with regard to seasonal dynamics (Fig. 8) and interannual variations (Fig. 9). Areas of ice age fractions of each age in the Arctic ocean are estimated individually from the NSIDC and SICCI products and plotted against time. For the NSIDC product the area of an ice age fraction is calculated as a number of all pixels corresponding





to this ice age fraction multiplied by the area of the pixel. Similarly the MYI area from the OSI SAF product is estimated from the number of MYI pixels. For the SICCI product the area is estimated as the integral of the corresponding ice age fraction over all pixels. Similarly the MYI area from the BU product is estimated as the integral of MYI concentration over all pixels.

The seasonal variations of ice age fraction areas follow similar pattern for the NSIDC and SICCI products (Fig. 8): the sea
ice area minimum ($4 \times 10^6$ km$^2$) is observed in mid September, it is followed by a rapid growth of the first year ice (blue color on fig. 8) until beginning of the next year. A stable winter, when total area remain practically unchanged at level of $7 - 8 \times 10^6$ km$^2$, is over in May and is followed by a rapid decrease of FYI. Unlike FYI the area of MYI fractions cannot increase - it only gradually decreases due to ice convergence and melting. Maximum area of the second year old ice (2YI) is also observed in mid September - at this point all FYI from the previous year is considered as the ice which has survived summer melting and
it becomes 2YI. Other fractions of older ice increase their age by one year in the same fashion.

The MYI area is shown on Figs. 8 and 9 as a demarcation of the blue and orange areas and as black and yellow dots. A comparison of MYI areas from the four products reveals several differences. The NSIDC MYI area decreases in two regimes: a relatively slow decrease during winter solely due to convergence and a faster decrease in the end of spring when melting reaches MYI. The SICCI MYI area decrease also accelerates towards spring but more gently - melting affects MYI already
starting from the beginning of spring. The MYI area from UB and OSI SAF correspond well to the SICCI algorithm but exhibits much more sporadic variability including short periods of reasonless increase in MYI area. Although the MYI extent in the OSI SAF product is somewhat lower than of the other products (see maps on Fig. 7) the OSI SAF MYI area corresponds well to other estimates (Figs. 8 and 9) because concentration of MYI within the ice extent is assumed to be 100%.

By the beginning of 2013 the SICCI product has only one fraction of MYI (2YI) because calculations were initiated in 2012
and all MYI was assigned to be 2YI. By the fourth annual cycle (in 2016) the SICCI product has 5 ice age fractions and the distribution can be better compared with the NSIDC product. Clearly, the fractions of the older ice in the SICCI product have lower area than in the NSIDC product.

The interannual variability can be seen on the plot of areas of the ice age fractions for the 1st January of 2013, 2014, 2015 and 2016 on Fig. 9. The total ice area (entire height of the columns) is lower for the SICCI product because it is calculated as
an integral ice concentration over all pixels whereas for the NSIDC it is calculated as sum of pixels within ice extent. The total MYI area is rather similar across the four products, but the area of older sea ice ($\geq$ 2years) fractions is lower by almost 20% in the SICCI product.





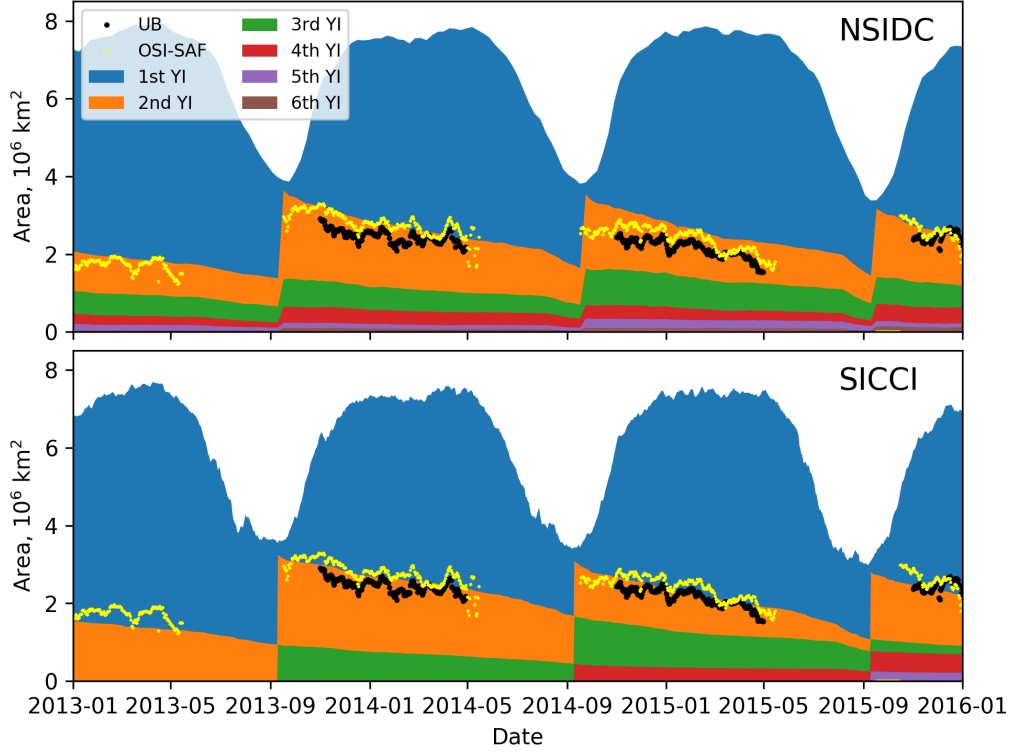

**Figure 8.** Seasonal dynamics of area of sea ice age fractions (filled plots) and multi-year ice area (dots) derived from the NSIDC product (upper part), from the SICCI product (lower part), from the UB product (black dots) and from the OSI SAF (yellow dots).

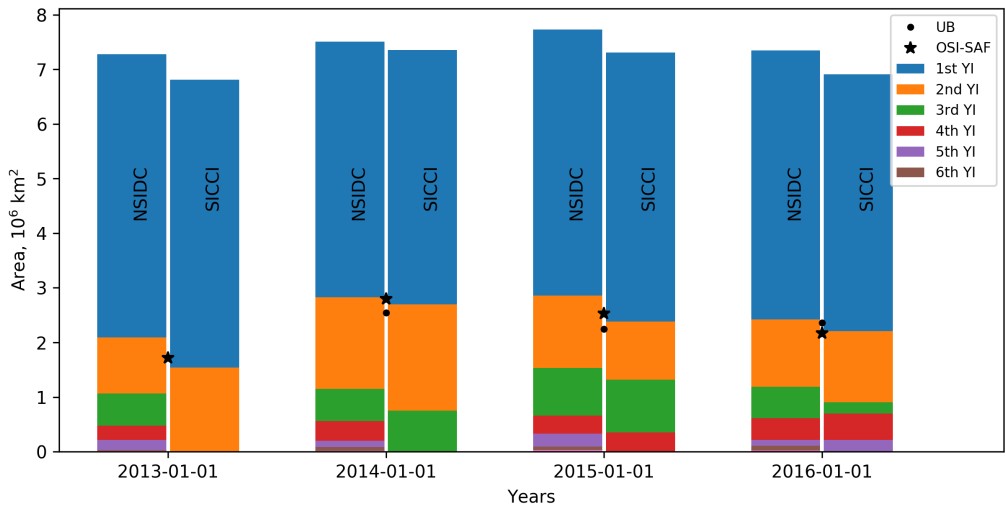

**Figure 9.** Interannual dynamics of area of sea ice age fractions (bars) and multi-year ice area (dots) derived from NSIDC product (left bars), from NERSC product (right bars), from UB product (black dots) and from OSI SAF (black stars).



## 4.3 Comparison of MYI with SAR

An independent validation of the SICCI MYI product was performed using a mosaic of Synthetic Aperture Radar (SAR) images in HH polarization from Sentinel-1 A and B satellites acquired around 1 January 2016. MYI appear as brighter and more homogeneous texture on SAR images which allows to draw an outline and compare it with outlines of MYI from the

5   SICCI product as shown on Fig. 10. The SICCI MYI product corresponds very well to the SAR derived MYI extent although it has much smoother boundaries due to low resolution of the input sea ice drift product (65 km). The low resolution is also the reason why SICCI MYI does not reach coast leaving a 50 km gap along the shoreline. The most considerable difference is observed in the Beaufort Sea where the SAR based MYI is seen as a stretched 'archipelago' of MYI 'islands'. In this area SICCI product exhibits erroneously smooth decline in MYI concentration and fills the gap between the MYI 'archipelago' and

10  the central Beaufort sea.

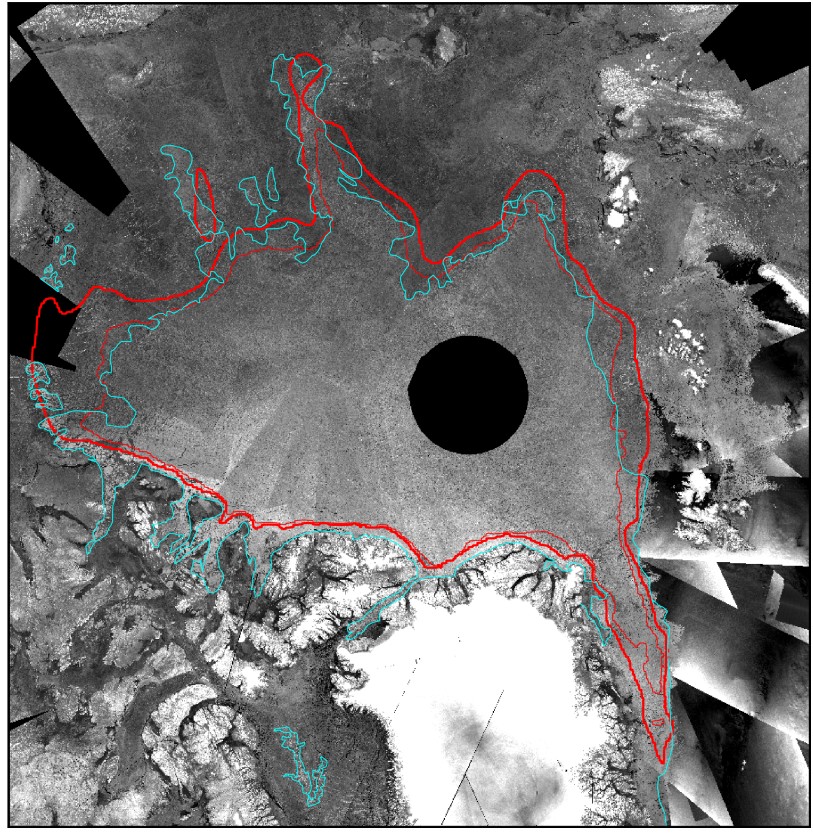

**Figure 10.** Comparison of MYI extent derived by SAR visual interpretation (light blue) and from SICCI SIA product (red) on top of mosaic of Sentinel-1 SAR images for 1 Jan 2016. The thick red line shows 15% threshold and thin red line shows 50% threshold.

## 5  Discussion

The major advantage of the new algorithm is the ability to generate individual ice age fractions or, in other words, to provide a frequency distribution of the ice age in each pixel. This allows derivation of a single number characterizing SIA using one of the statistical methods presented above. Selection of the method depends on the preferred definition of the sea ice age and on

the eventual application. For example, in our opinion weighted average is the optimal method for graphical representation of SIA maps as it depicts the smooth nature of SIA distribution and accounts not only for the oldest ice fractions but also for the younger ones. However, for other applications, e.g. assimilation into models, a different method may be preferred. The SIA distribution can also be used to calculate MYI concentration and use it for many purposes including: calibration/validation of the PAMW-based MYI algorithms; improvement of the freeboard to sea ice thickness conversion for altimeter data; estimation

of sea ice roughness for assimilation into models.

The motivation for implementing the Eulerian advection scheme in the new algorithm was to produce continuous and smooth spatial distributions of sea ice age fractions and also to prevent undersampling of the results. In the experiments with the NSIDC algorithm it was discovered that the density of starting points of the sea ice drift vectors indirectly influences the area of MYI. Too low density (e.g. 1 drifter in each $12 \times 12$ km box as designed at NSIDC) leads to undersampling of the ice age observations

in the zone of strong divergence very early - only after a few years of propagation. Although it happens later - maybe after 5 years of propagation - but a higher number of initial points still leads to undersampling when the Lagrangian propagation scheme is used. The NSIDC method was run with three different initial densities of ice parcels: one drifter per $12 \times 12$ km or $6 \times 6$ km or $3 \times 3$ km box. The results show that lower initial densities result in a map of SIA with sparse presence of old ice fractions (Fig. 11, A) and increasing the initial density of drift vectors leads to increase of concentration of the older ice

(Fig. 11, B and C). Consequently the area of older ice fractions $\geq$ 6 years) increases and the area of younger ice fractions ($\leq$ 5 years) decreases when the initial density is increased.



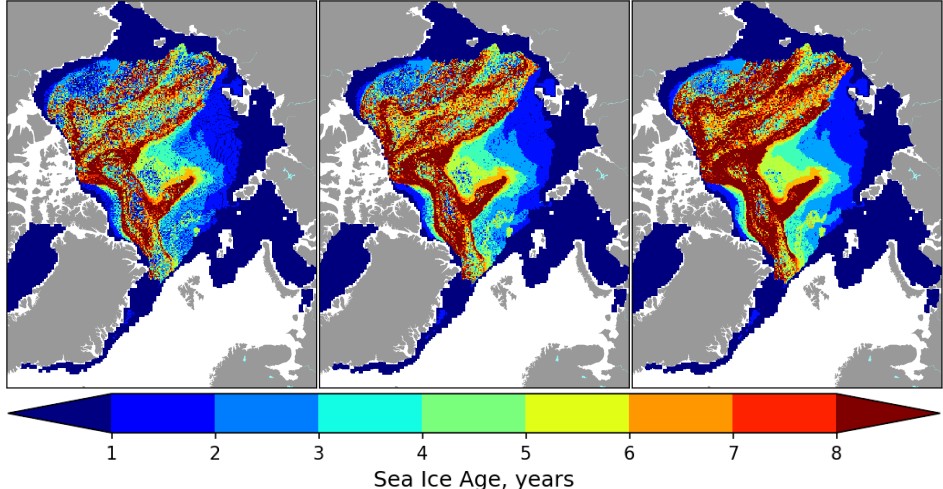

**Figure 11.** Comparison of sea ice age for 1 Jan 1985 initiated from ice parcels with various density (1 parcel per: 12 x 12 km, 6 x 6 km, 3 x 3 km).

The new sea ice drift product derived from AMSR2 also has a significant impact on the accuracy and reliability of the results. It accurately reproduces the sea ice circulation in the Arctic, it is void of artifacts and produces homogeneous distribution of sea ice age fractions. It was discovered that the initial SID product is contaminated with high frequency noise. Cleaning of SID with a median filter reduced small scale deformations and prevented appearance of FYI speckles in artificial divergence zones.

At the same time the low spatial resolution of SID (65 km) limits the algorithm from reproduction of fine scale features in MYI distribution and may lead to over-smoothing in areas with high ice drift speed gradients (e.g. western Beaufort sea).

The observed differences in MYI spatial distribution and total area are only marginal for the NSIDC and the SICCI products. This happens due to a short integration time: MYI spatial distribution is forced by observations on 15 September and then it is modified by an advection scheme (either Lagrangian or Eulerian) and melting (in case of SICCI product) during only one

year. On the contrary, the integration over longer times and accounting for the sea ice concentration have significant effect on the area of the older ice. Analysis of inter-annual dynamics (Fig. 9) shows that after 4 years of integration FYI and MYI areas in the NSIDC and SICCI products are almost the same ($7.1 \times 10^6$ km$^2$ and $6.9 \times 10^6$ km$^2$) but the area of the $3^{rd}$-year and older ice in the SICCI product is almost 30% smaller that in the NSIDC product. On average, the speed of annual multi-year ice decline (calculated as difference between MYI area in the beginning and in the end of the year) is 22% higher for the SICCI

product which may also indicate that the actual residence time of sea ice is also shorter by one fifth. More accurate estimates of the older sea ice dynamics and long term changes in the residence time are only possible when longer time series of the correct ice drift product will be available.

The sea ice type algorithms based on PAMW satellite observations have higher spatial resolution than the ice drift based algorithms but inevitably suffer from the unaccounted atmospheric impact or melt ponds and have to be corrected using

information on air temperature or ice motion (Ye et al., 2016a), (Ye et al., 2016b). The method in our paper has high potential for complementing the PMW algorithms and production of high resolution and consistent MYI estimations. Such combined





procedure can be realized, for example, as a machine learning algorithm which is trained to output SICCI MYI using PMW brightness temperatures ($T_B$) on input. The machine learning block can be realized as a simple polynomial regression, or as a more complex neural network (Haykin, 1998). The training can be performed either on a per image basis, when only one mosaic of brightness temperatures is used as input and only one snapshot of MYI is used as output, or on seasonal (or even

satellite mission) basis, when information from many images is used for training. Alternatively, both SICCI MYI estimations and brightness temperatures can serve as input to classification algorithms, such as Bayesian classifier (Zhang, 2004) or Support Vector Machines (Smola and Schölkopf, 2004).

## 6   Conclusions

We have developed a new algorithm for estimating sea ice age distribution using sea ice drift and concentration products.

The algorithm is based on the Eulerian advection scheme which provides smooth distribution of the ice age parameters and prevents the undersampling problem that may occur when a Lagrangian tracking approach is used. Another advantage of the selected scheme is the ability to generate not just a single age characteristic but a distribution of sea ice age fractions. First, this allows for flexibility in choosing the ice age definition and application of a statistical measure to compute SIA and, second, this provides individual spatial distributions of ice age fractions that can be assimilated into models or used for ice type delineation.

For example, concentration of multiyear ice can be computed as a sum of multi-year ice fractions and used for defining ice density and snow thickness for the ice thickness algorithms, ice roughness for the ice circulation models and so on.

The new algorithm is driven by the new sea ice drift products from OSI SAF which is void of potential artifacts due to inclusion of autonomous ice drifter buoys. This leads to a more homogeneous distribution of ice age fractions over the Arctic Ocean. The algorithm is also constrained by the observed sea ice concentration from OSI SAF which reduces fractions of old

ice and, consequently, ice age by 20 – 30%. It was applied to generate time series of daily sea ice age fraction product starting from October 2012 to October 2017. Comparisons with the NSIDC SIA time series indicate that the fractions of MYI in the new product melt faster during the year and after spin-off time of three years the area of older ice in the SICCI product is almost 20% lower than in the NSIDC product.

*Data availability.* The data generated with the algorithm is openly available at FTP (for bulk download): ftp://ftp.nersc.no/ArcticData/esa_

cci_sea_ice_age/ and at THREDDS (subsetting and online visualization): http://thredds.nersc.no/thredds/arcticData/esa-cci-sea-ice-age.html in netCDF format following CF-conventions containing values of sea ice age fractions concentrations, MYI concentration, and sea ice age computed using weighted average.

*Author contributions.* AK developed and applied the presented algorithm, with contributions from PR. TL provided the new sea ice drift product. SA provided the OSI SAF ice type data. YY and GH provided the new sea ice type product. LTP and RS provided the mosaic of

SAR images and multiyear ice outline. All co-authors participated in fruitful discussions and writing the manuscript.





*Competing interests.* The authors declare that they have no conflict of interest.

*Acknowledgements.* This work has been supported by the Sea Ice Age option of the Sea Ice Climate Change Initiative project funded by the European Space Agency, contract number 4000112229/15/I-NB.



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
