# Peer review of "A new tracking algorithm for sea ice age distribution estimation"

_The Cryosphere, 2017_

## Referee Comment (RC1) · Anonymous Referee #1 · 20 Dec 2017

This paper introduces a new method to calculate ice age. It is well-written, sound in its approach and produces scientifically interesting results.

Specific comments:

p.,Line # Comment

p.2, 30* The Szanyi et al. 2016 paper identified a known problem with buoy influence on drift speed, which has since been fixed in the current sea ice motion and age Version 3 datasets at NSIDC, rendering their paper invalid, other than as a historical look at an outdated dataset.

P. 7, L 18 Of course, postulating that all observed ice is 2nd-year is erroneous. The NSIDC ice age algorithm initiates ice as first-year, then increments by one year after

the minimum extent is reached. But the dataset is not considered mature until 5 years out, using a 5+ age maximum category. Granted, AMSR-2 began in 2012, but it's probably not too informative to compute ice age with it until 2017.

P,17, 2 Yes, a distribution of ice age fraction within a cell is a sound improvement, more accurately representing the ice age within that cell.

---

## Referee Comment (RC2) · Anonymous Referee #2 · 7 Mar 2018

General comments

The manuscript "A new tracking algorithm for sea ice age distribution estimation" by A. Kosorov, P. Rampal, L. Pedersen, R. Saldo, Y. Ye, G. Heygster, T. Lavergne, S. Aaboe, F. Girard-Ardhuin, provides a comprehensive analysis of techniques and data used to estimate sea ice age distributions. This paper provides a clear description of a new algorithm and product that results in improved characterization and understanding of sea ice age distributions. In addressing issues associated with unrealistic ice divergence and convergence resulting from artifacts in the NSIDC sea ice drift field, in addition to limitations associated with existing algorithms used to estimate sea ice age, this paper shows that the OSISAF sea ice drift product combined with the SICCI algorithm provides a reasonable and improved alternative to existing data products and algorithms

traditionally used to characterize sea ice age.

The paper addresses in a comprehensive and coherent manner relevant scientific questions associated with sea ice age and its characterization. In addition, the manuscript builds on and presents novel concepts and ideas that contribute to an understanding of sea ice dynamics and implications for sea ice age distribution estimation relevant for data assimilation, validation, and machine learning development applications. Substantial conclusions, including the development of a product (OSISAF data and Eulerian tracking algorithm) to address inconsistencies in, and provide additional information beyond, what is provided in current data products, including sea ice age distributions at each pixel, are found. Furthermore, scientific methods and assumptions are clearly presented, including the Eulerian advection scheme with emphasis on advected sea ice age fractions, and ice age probability distributions. Interpretations of results and conclusions are supported by comparison between four scenarios and products, and with SAR-derived MYI extent. Additional reference to, and description of, existing studies that support the results from this analysis, including undersampling issues and an under-estimation of MYI sea ice extent using the NSIDC data product and Lagrangian approach, could be provided in the Introduction and motivation for the present study. Although the title accurately reflects the manuscript objectives, text within the abstract could be clarified to reinforce study findings and highlight the benefits of this approach for improved understanding of sea ice age distributions and MYI ice extent relevant for validation and assimilation applications. The paper is well structured, and formulae and figures clear. This is to recommend that the manuscript be accepted for publication with minor edits. Please find below more specific comments for consideration.

Specific comments

Abstract

p. 1, lines 7 − 8. "Several improvements related to the usage of the new ice drift

product, constraining the observed algorithm by the ice concentration and preventing undersampling by the Eulerian scheme are presented." Perhaps replace this statement with 2 – 3 phrases highlighting the key findings of the study, including the i) development of a new sea ice age product that combines OSISAF ice drift data with an Eulerian algorithm, ii) derivation of sea ice age distributions based on the additional constraint of sea ice concentrations, and iii) more realistic representation of MYI that addresses under sampling found using the standard Lagrangian approach and NSIDC ice drift product to characterize sea ice age.

Introduction

Additional studies that provide motivation and context for the present analysis could be included in the Introduction, with the paper by Rybak and Hubrechts (2003) as an example.

Data

p. 2, line 32. In response to the first reviewer's comment, perhaps note that earlier studies showing persistent artifacts also used version 3 of the NSIDC sea ice drift product.

Methodology

p. 5, line 5. Perhaps indicate the impact of uncertainty in ice drift and concentrations on the Eulerian advection scheme, in addition to pixel size relative to the floes considered.

p. 7, line 12. How do results differ for a different start date?

p. 7, line 19. 'We postulated that all observed ice is second year ice'. What are the implications of this assumption and an initially spatially homogeneous ice age distribution?

p. 8, line 10. It would be beneficial to compare these diagnostics with those used for the standard Lagrangian NISDC sea ice age algorithm, and to highlight potential applications for each, as is noted briefly in the Discussion. In particular, when should the median rather than the weighted average age be used for sea ice age characterizations? For what applications would the maximum, average, and modal age diagnostics and interpretations be appropriate?

Results

p. 10, line 7. It should perhaps be noted when comparing the NSIDC and SICCI products that the SIA produced by NSIDC assigns the age of the oldest particle to that cell (Maslanik et al., 2011). In addition, panels A and B in Figure 6 show that NSIDC forcing underestimates MYI extent in contrast to OSISAF forcing using the Lagrangian approach, in keeping with Szanyi et al. (2016).

p. 10, line 19. Perhaps include 'for OSISAF relative to NSIDC ice drift forcing' at the end of this sentence.

p. 12, line 2. Perhaps replace "speckles" with "discontinuities".

p. 12, line 14. 'SICCI SIA product' Is sea ice age defined according to the weighted product and consideration of SIC?

p. 12, line 20. "speckles" could be replaced with "discontinuities", as previously noted. Similarly throughout the manuscript.

p. 12, line 20. Perhaps highlight that lacuna and gaps in the sea ice age maps are associated with openings that are filled with FYI, as described on p. 3, and relate this to persistent artifacts in the NSIDC ice drift field associated with incorporation of buoy data. Panels depicting NSIDC sea ice drift fields for the indicated dates could be included in the first row of Figure 7.

Discussion and conclusions

p. 17, line 2. First statement. This is a key finding of the present study, and should be reiterated and included in the Abstract.

p. 17, line 14. 'Too low sampling. . .', as is also noted in Szanyi et al., 2016.

p. 17, line 18. 'The results show that. . .' Is the NSIDC or OSISAF ice drift data used for this analysis? It might be helpful to replicate these panels using the OSISAF dataset to determine whether similar patterns emerge with increased sampling using the Lagrangian approach.

p. 18, line 14. Differences in residence times may also be due to different ice drift products, in addition to differences associated with Lagrangian and Eulerian algorithms.

p. 19, line 11. Are undersampling issues not also a result of inconsistencies and persistent features in the NSIDC dataset?

The sea ice age product presented in the current study provides a novel alternative to existing sea ice age products through its spatial characterization of sea ice age distributions, with a number of potential applications, and implications for our understanding of relative changes in MYI and FYI in the Arctic.

Technical corrections

p. 1, line 17. Replace 'been focusing' with 'focused '.

p. 2, lines 8. Replace 'deform' with 'deformation'.

p. 4, line 1. Please clarify the phrase 'developed to apply on'.

p. 9, Figure 5 caption. The description for panels c) and d) should be reversed.

p. 10, line 4. 'DTU' Please clarify.

p. 12, line 26. Please change 'seem' to 'seems'.

p. 14, line 16. 'reasonless'. Please clarify. Perhaps this could be replaced with 'an inexplicable'?

p. 16, line 7. Please insert 'the' between 'reach' and 'coast'.

p. 18, line 14. Please replace 'that' with 'than'.

Reference

Rybak, O., and P. Huybrechts, 2003: A comparison of Eulerian and Lagrangian methods for dating in numerical ice-sheet models, Annals of Glaciology, 37.

Please also note the supplement to this comment:
https://www.the-cryosphere-discuss.net/tc-2017-250/tc-2017-250-RC2-supplement.pdf

---

## Author Comment (AC1) · 22 Mar 2018

**Replies to the Anonymous Referee #1**

(the referee comments are in blue, our replies in black)

We are grateful for the positive review of our manuscript.

**Specific comments:**
p.,Line # Comment
p.2, 30* The Szanyi et al. 2016 paper identified a known problem with buoy influence on drift speed, which has since been fixed in the current sea ice motion and age Version 3 datasets at NSIDC, rendering their paper invalid, other than as a historical look at an outdated dataset.
In their paper Szanyi et al., have evaluated the product "Polar Pathfinder Daily 25 km EASE-GRID Sea Ice Motion Vectors, Version 3". We have also used this product as that was the latest available dataset from NSIDC. And we have also identified that at least in the beginning of the passive microwave remote sensing era some artifacts are observed (see fig. 1 below). These artifacts potentially arise due to assimilation of information from drifting buoys and obviously increase coverage by first year ice (via increased divergence). We decided not to include this figure in the manuscript in order to focus on the new results.

[Figure]

**Figure 1**. Maps of first-year ice (dark blue) and multi-year ice (light blue) derived with the Lagrangian tracking approach (Maslanick et al., 2011) from the NSIDC sea ice motion product, version 3 for 22 October 1979. Black dots show position of several IABP buoys for this date, red dots - their trajectories for the last 30 days.

P. 7, L 18 Of course, postulating that all observed ice is 2nd-year is erroneous. The NSIDC ice age algorithm initiates ice as first-year, then increments by one year after the minimum extent is reached. But the dataset is not considered mature until 5 years out, using a 5+ age maximum category. Granted, AMSR-2 began in 2012, but it's probably not too informative to compute ice age with it until 2017.

Indeed, it is more correct to postulate that all observed ice is at least in the 2nd-year category. However, we have to allow a simplification that all observed ice is 2nd-year ice. This simplification is sufficiently accurate for two purposes, first, for initialization of the ice age product computation and, second, for explanation of the ice age algorithms in the manuscript. The sentences are changed accordingly:

*We did not know the spatial distribution of ice of different age within the pack at the first moment of time (1 Oct 2012), but we can postulate that all observed ice is at least in the second year ice (2YI) category. We have to make a simplification: the concentration of second year ice on 1 Oct 2012 is assigned to be equal to the total observed concentration: $C_{2YI} = C_{TOT}$ (Fig. 3, A). Understandingly, this simplification allows to initialize the sea ice age algorithm but does not allow to provide full sea ice age frequency distribution before a sufficiently long spin up period (e.g. 5 years). Such a spin up period is also needed for the NSIDC product.*

P,17, 2 Yes, a distribution of ice age fraction within a cell is a sound improvement, more accurately representing the ice age within that cell.
Thank you!

**Replies to the Anonymous Referee #2**

(the referee comments are in blue, our replies are in black)

We are grateful for a comprehensive review of our manuscript and detailed recommendations for improvements.

**Specific comments**
**Abstract**
p. 1, lines 7 – 8. "Several improvements related to the usage of the new ice drift product, constraining the observed algorithm by the ice concentration and preventing undersampling by the Eulerian scheme are presented." Perhaps replace this statement with 2 – 3 phrases highlighting the key findings of the study, including the i) development of a new sea ice age product that combines OSISAF ice drift data with an Eulerian algorithm, ii) derivation of sea ice age distributions based on the additional constraint of sea ice concentrations, and iii) more realistic representation of MYI that addresses under sampling found using the standard Lagrangian approach and NSIDC ice drift product to characterize sea ice age.
The aforementioned sentence is replaced with the following text:
*Comparison with the National Snow and Ice Data Center SIA product revealed several improvements of the new SIA maps and time series. First, application of the Eulerian scheme provides smooth distribution of the ice age parameters and prevents product undersampling which may occur when a Lagrangian tracking approach is used. Second, utilization of the new, void of artifacts sea ice drift product from EUMETSAT OSI SAF resulted in more accurate and reliable spatial distribution of ice age fractions. Third, constraining SIA computations by the observed sea ice concentration expectedly led to considerable reduction of multi-year ice fractions.*

**Introduction**
Additional studies that provide motivation and context for the present analysis could be included in the Introduction, with the paper by Rybak and Hubrechts (2003) as an example.
The sentence in the Introduction is rewritten as follows:
*These improvements have allowed us to avoid the problem of the tracers dispersion (Rybak and Hubrechts, 2003) and to produce a new sea ice age dataset which in each grid box contains not only the age of the oldest ice, but the actual age distribution provided as fractions of ice of different age categories (hereafter referred to as sea ice age fractions).*
The following sentence was added to the Discussion:
*Observations that the Lagrangian approach introduces problems due to the dispersion of tracers were also previously reported by Rybak and Huybrechts (2003).*

**Data**
p. 2, line 32. In response to the first reviewer's comment, perhaps note that earlier studies showing persistent artifacts also used version 3 of the NSIDC sea ice drift product.

This is noted and an example of the artifacts effect on sea ice age distribution is provided to reviewer 1.

**Methodology**

p. 5, line 5. Perhaps indicate the impact of uncertainty in ice drift and concentrations on the Eulerian advection scheme, in addition to pixel size relative to the floes considered.

The following sentence is added:

*Uncertainties in U, V and $C_{OBS}$ products impact the accuracy of the end product but are not considered in the present study.*

There is no relation between the size of the actual ice floes and pixels.

p. 7, line 12. How do results differ for a different start date?

The initialization of the algorithm before the observed ice area minimum, has little impact: FYI will not appear or will melt anyway. The initialization after the minimum has negative impact: large areas with FYI will be considered as MYI. In fact, in our experiments we initialized from the map of minimum sea ice concentration over the period 1 Sep - 1 Oct 2002.

The section is slightly rewritten:

*Advection of a sea ice age fraction is initiated on 10 September, the approximate date when ice extent reaches minimum in the Arctic, area of first-year ice (FYI) is zero and all observed sea ice is multi-year ice (MYI) by definition. Initialization before this date has little impact, but initialization after this date increases risk to consider the observed FYI as MYI. The age of each ice fraction is increased by one year also on 10 September of each consecutive year of advection.*

*In our study we initiated SIA calculation on 1 October 2012 when continuous high quality observations of ice drift started to be available from AMSR2 (Fig. 3). For this date the total observed concentration was computed as the minimum concentration during the period 1 September - 1 October 2002. Propagation of ice age fractions from later years started from 10 September or respective years.*

p. 7, line 19. 'We postulated that all observed ice is second year ice'. What are the implications of this assumption and an initially spatially homogeneous ice age distribution?

The main implication is that the sea ice age distribution in a pixel will not be complete until the algorithm is run for a sufficient spin-off period. During the first 4 - 5 years of computations we can distinguish only younger fractions (e.g. FY1, 2YI, 3YI) and sum of older fractions (e.g. at-least-4-years-old-ice). The sentence is rewritten also per request from the first reviewer:

*We did not know the spatial distribution of ice of different age within the pack at the first moment of time (1 Oct 2012), but we can postulate that all observed ice is at least in the second year ice (2YI) category. We have to make a simplification: the concentration of second year ice on 1 Oct 2012 is assigned to be equal to the total observed concentration: $C_{2YI} = C_{TOT}$ (Fig. 3, A). Understandingly, this simplification allows to initialize the sea ice age algorithm but does not allow to provide full sea ice age frequency distribution before a sufficiently long spin up period.*

 It would be beneficial to compare these diagnostics with those used for the standard Lagrangian NSIDC sea ice age algorithm, and to highlight potential applications for each, as is noted briefly in the Discussion. In particular, when should the median rather than the weighted average age be used for sea ice age characterizations? For what applications would the maximum, average, and modal age diagnostics and interpretations be appropriate?

SIA map from the NSIDC product was added to the figure. Fig. 5 is just an illustration how ice age distribution varies depending on the definition and is not meant to be a guidance for usage in specific applications.

**Results**

 It should perhaps be noted when comparing the NSIDC and SICCI products that the SIA produced by NSIDC assigns the age of the oldest particle to that cell (Maslanik et al., 2011). In addition, panels A and B in Figure 6 show that NSIDC forcing underestimates MYI extent in contrast to OSISAF forcing using the Lagrangian approach, in keeping with Szanyi et al. (2016).

Several sentences are rewritten as follows:

*First, we have implemented the NSIDC advection scheme (Lagrangian) and the SIA algorithm (age of the oldest parcel) and applied it to the ice drift product from NSIDC. Second, we have applied the NSIDC algorithms to the ice drift from OSI SAF.*

*The change of the forcing significantly affects distribution of SIA (Fig. 6, A and B) as also illustrated by Szanyi et al. (2016).*

 Perhaps include 'for OSISAF relative to NSIDC ice drift forcing' at the end of this sentence.

The sentence is changed accordingly.

 Perhaps replace "speckles" with "discontinuities".

"Speckles" are replaced with "discontinuities" throughout the text.

 'SICCI SIA product' Is sea ice age defined according to the weighted product and consideration of SIC?

Yes, SIC was considered. But no weighted averaging of ice age fractions was performed for the purpose of MYI calculation. Instead, fractions of MYI were summed up. The sentence is rewritten as follows:

*Total concentration of MYI from the SICCI product (with due consideration of SIC) was estimated as a sum of all multi-year fractions.*

 "speckles" could be replaced with "discontinuities", as previously noted. Similarly throughout the manuscript.

"Speckles" are replaced with "discontinuities" throughout the text.

 Perhaps highlight that lacuna and gaps in the sea ice age maps are associated with openings that are filled with FYI, as described on p. 3, and relate this to persistent artifacts

in the NSIDC ice drift field associated with incorporation of buoy data. Panels depicting NSIDC sea ice drift fields for the indicated dates could be included in the first row of Figure 7.

Neither a single snapshot nor an averaged ice drift field allow to easily identify the artifacts. Only integration of trajectories over some time (at least few weeks) reveals impact of these artifacts on ice age. Therefore, we believe it will not be illustrative to include maps with NSIDC ice drift into the figure.

The phrase is rewritten as follows:

*…; numerous discontinuities due to artifacts in the sea ice drift field are present; ...*

**Discussion and conclusions**

p. 17, line 2. First statement. This is a key finding of the present study, and should be reiterated and included in the Abstract.

The sentence in the Abstract is rewritten as follows:

*The major advantage of the new algorithm is the ability to generate individual ice age fractions in each pixel of the output product or, in other words, to provide a frequency distribution of the ice age allowing to apply mean, median, weighted average or other statistical measures.*

p. 17, line 14. 'Too low sampling...', as is also noted in Szanyi et al., 2016.

The sentence is rewritten accordingly.

p. 17, line 18. 'The results show that...' Is the NSIDC or OSISAF ice drift data used for this analysis? It might be helpful to replicate these panels using the OSISAF dataset to determine whether similar patterns emerge with increased sampling using the Lagrangian approach.

The following sentence is added:

*Impact of the increased initial density on increased MYI area is independent of the forcing and was demonstrated also for the OSI SAF ice drift field (not shown here).*

p. 18, line 14. Differences in residence times may also be due to different ice drift products, in addition to differences associated with Lagrangian and Eulerian algorithms.

The sentence is rewritten as follows:

*It is challenging to separate the impact of the new sea ice drift product, scheme of advection and constraining by the SIC but more accurate estimates of long term changes in the residence time will become possible when longer time series of the new SIA product will be available.*

p. 19, line 11. Are undersampling issues not also a result of inconsistencies and persistent features in the NSIDC dataset?

No, undersampling is the feature of the Lagrangian approach and occurs also when OSI SAF drift is used as forcing. This is now mentioned in Discussion (see above).

The sea ice age product presented in the current study provides a novel alternative to existing sea ice age products through its spatial characterization of sea ice age distributions, with a

number of potential applications, and implications for our understanding of relative changes in MYI and FYI in the Arctic.

This sentence is added to conclusions.

**Technical corrections**

p. 1, line 17. Replace 'been focusing' with 'focused '.

p. 2, lines 8. Replace 'deform' with 'deformation'.

p. 4, line 1. Please clarify the phrase 'developed to apply on'.

p. 9, Figure 5 caption. The description for panels c) and d) should be reversed.

p. 10, line 4. 'DTU' Please clarify.

p. 12, line 26. Please change 'seem' to 'seems'.

p. 14, line 16. 'reasonless'. Please clarify. Perhaps this could be replaced with 'an inexplicable'?

p. 16, line 7. Please insert 'the' between 'reach' and 'coast'.

p. 18, line 14. Please replace 'that' with 'than'.

All technical corrections are implemented.

**Reference**

Rybak, O., and P. Huybrechts, 2003: A comparison of Eulerian and Lagrangian methods for dating in numerical ice-sheet models, Annals of Glaciology, 37.

---

## Referee Report (RR1)

This study presents new method of Eulerian advection scheme to estimate the sea ice age. The paper is well-written and the figures are of high quality. The method and derived SIA is very powerful and will make a great contribution to the studies of sea-ice dynamics, Arctic climate system, etc. I think this paper is suitable for publication in TC.

Comments:

P5 L8: As a reviewer pointed out, uncertainties in ice drift speed and ice concentration have an impact on the ice age estimation. But I understand the author's giving up on the assessment of the errors arising from the uncertainties, because the error assessment is very difficult.

P7 L9: In the case of $C_{oi} > C_{obs}$, this study assume that $C_{oi} - C_{obs}$ to be ice melting. But ice thickening by ridging or rafting also occurs in the melting season. Can you separate the decrease of $C_{oi}$ by ice deformation from that by ice melting?

---

## Author Response (AR2)

Dear Jenny,

Thank you very much for your kind feedback and the suggestions! We have corrected the manuscript accordingly and done the proof reading.

Best regards!
Anton, Pierre, Leif, Roberto, Yufang, Georg, Thomas, Signe and Fanny

page 3, line 31: A suggestion to reword "The initial MYI concentrations are corrected by two correction schemes": Two correction schemes were applied to initial MYI concentration. This is a purely stylistic suggestion.
**Reply**: Corrected.

page 4, line 23: "Such case" -> "Such a case"
**Reply**: Corrected.

page 5, line 2: "details below" -> "detail below"
**Reply**: Corrected.

page 7, line 15: "increases risk to consider" -> "increases risk of considering".
**Reply**: Corrected.

page 7, line 16: "also" is not required in this sentance
**Reply**: Corrected.

page 10, line 2: "The developed advection scheme". Reference the section or paper where this is introduced.
**Reply**: Reference added.

page 10, line 12-13: "Finally, we have used the new advection scheme, the OSI SAF ice drift and fully accounting for SIC." Rephrase, this sentance does not make complete sense. Should the "and" be removed?
**Reply**: This sentence was rephrased the following way: *Finally, we have used the new advection scheme, the OSI SAF ice drift, and fully accounted for SIC.*

page 16, line 4: " allows to draw an outline". There are several examples of grammar mistakes like this throughout the paper. You could either phrase this as "allows one to draw an outline" or "allows drawing of an outline".

**Reply**: This sentence was rephrased the following way: *… which allows one to draw an outline and compare it with outlines of MYI from the SICCI product as shown on ….* Another occurrence was rephrased the following way: *… this simplification allows us to initialize the sea ice age ...*

page 17, line 15-17: Consider rephrasing " Although it happens later - maybe after 5 years of propagation - but a higher number of initial points still leads to undersampling when the Lagrangian propagation" scheme is used." Perhaps "With a higher number of initial points, undersampling is still experienced after 5 years of Lagrangian propagation". Also, it would help to quantify what "higher number of points" is.

**Reply**: This sentence was rephrased the following way: *With a twice or four times higher number of initial points, undersampling is still experienced after 5 years of Lagrangian propagation.*